# A STEPPED SAMPLING METHOD FOR VIDEO DETECTION USING LSTM

## ABSTRACT

Artificial neural networks are considered to simulate the human neural networks, and achieves great progress on object detection, natural language processing (NLP), image generation, etc. Hermann Ebbinghaus proposed the law of human memory and how to improve human learning in 1885. Inspiring from Ebbinghaus' work, we propose a stepped sampler based on the "repeated input", which is Ebbinghaus' contribution that how to strengthen the learning. We repeatedly inputted data to the LSTM model stepwise in a batch. The stepped sampler is used to strengthen the ability of fusing the temporal information in LSTM. We tested the stepped sampler on the LSTM offered by PyTorch. Compared with the traditional sampler of PyTorch, such as sequential sampler, batch sampler, the training loss of the proposed stepped sampler converges faster in the model training, and the training loss after convergence is more stable, which means that there is no large jitter after the convergence. Meanwhile, it can maintain a higher test accuracy, compared with the traditional sampler. We quantified the algorithm of the stepped sampler. We assume that, the artificial neural networks may have human-like characteristics, and human learning method could be used for machine learning. Our code will be available online soon.

## 1 INTRODUCTION

The emergence of convolutional neural networks (CNN) (LeCun et al., 1989) has improved the self-learning ability of artificial neural networks. Recurrent Neural Network (RNN) (Mikolov et al., 2010) is used to process the temporal information data. RNN takes the output of the previous time period as the input of the next time period, effectively using the temporal information of the input sequence.

RNN sometimes may have the problem of gradient disappearance or gradient explosion. Hochreiter et al. (Hochreiter & Schmidhuber, 1997) proposed LSTM. LSTM adds gates to RNN, thus it can effectively avoid the problem of gradient disappearance or explosion. These gates include the forgetting gates, the input gates, and the output gates. The forgetting gate seems to be the most important among them. LSTM may simulate the memory process of human brain. Human brain selectively forgets some information for learning better.

Consider that one of the principles of neural networks may be learned from biological neural networks, for those artificial neural networks with the memory effects, such as LSTM, learning from the memory method of human, which is the repeated input and timely review, we study the effect of this method with repeated input on LSTM detection results, without considering changing the LSTM network structure.

In this study, we learn the effect of the proposed input method on neural networks with memory characteristics, such as LSTM. Specifically, it is to repeatedly input training data by simulating the "repeated input and timely review" method of the human memory, and the "repeated input and timely review" method is proposed by Hermann Ebbinghaus (Ebbinghaus, 1913) in 1885, which is the "Increasing Memory Strength and Rate of Learning" in his literature.

## 1.1 OUR CONTRIBUTION

Our views in this paper mainly include the following 3 aspects:

**a)** A novel sampler is proposed, which implements sampling in a circular and stepwise manner. Compared with the traditional sampler, the loss curve of the LSTM model using this stepped sampler converges faster in training, and is more stable after the convergence, namely there is no large jitter after the convergence. Moreover, its test accuracy curve is more stable either, which has no jitter. When the batch size is 15, the test accuracy of the stepped sampler LSTM is much higher than that of the traditional sampler with the same parameters.

**b)** The idea of this sampler comes from the laws of human memory, which was proposed by Ebbinghaus (Ebbinghaus, 1913). We courageously assume that, other human learning methods can also be applied to machine learning. One example is the proposal of the attention mechanism (Vaswani et al., 2017). Moreover, we believe that artificial neural networks have human-like characteristics from the experimental performance.

**c)** We try to use mathematical language to describe the temporal information of the video frames. We try to apply the mathematical equations to our experimental results, and analyze that the test accuracy in the experiment is the temporal information between video frames. The derivation process is shown in Appendix A and Appendix B.

## 2 RELATED WORK

Gibbs sampling is one of the earlier data sampling algorithms, which is proposed by Geman et al. (Geman & Geman, 1984) in 1984. Gibbs sampling is to make the probability of the data sample approximately equal to the required probability distribution via iterations. Gibbs sampling randomly selects data from an initial input sequence, and iterates according to the specified conditional probabilities, which are related to the required probability distribution of the final sampling data. After iterations, Gibbs sampling generates data which is consistent with the required probability distribution. Hu et al. (Hu et al., 2018) used neural networks to generate a sampler, which transfer the initial data distribution to the target distribution. The method can generate the sampling data at the same time of training. This method works with the un-normalized probability density function. Wang et al. (Wang et al., 2018) used Generative Adversarial Nets (GAN) (Goodfellow et al., 2014) to generate the negative samples. The approach is the first to combine GAN with the negative sampling method, which improves the training effect of the streaming recommend system. Chu et al. (Chu et al., 2019) proposed a novel sampler that can sample both the positive and the negative data from the input data sequences, so as to let the classifier utilize the Regions of Interests and the background of the data. The sampler is used in the few-shot image classifier, which uses the reinforcement learning method. The reinforcement learning algorithm (Kaelbling et al., 1996) needs to continuously select the regions of interests from the images, subsequently to recognize the content of the Regions of Interests. Sampling these Regions of Interests can improve the efficiency of reinforcement learning, for the reason of the reduction of the input samples. Muhammad et al. (Muhammad et al., 2021) proposed a bi-directional long short-term memory (BiLSTM) with attention mechanism and a dilated convolutional neural network (DCNN) to perform action recognition, which outperformed the state-of-the-art methods. Kwon et al. (Kwon et al., 2021) proposed a spatio-temporal neighbourhood learning method on action recognition, which performed the state-of-the-art.

## 3 MATERIALS AND METHODS

This paper is from the perspective of data input, rather than the neural network structure, and study the impact of the memory effect on the temporal sequence neural networks (such as LSTM). The process simulates the method of enhancing the memory process of human brain, repeats the input data in a stepped way. The method is proposed by Hermann Ebbinghaus (Ebbinghaus, 1913) called "Increasing rate of learning" in his book. The specific mode we used was the wheel tactic (Smith, 1994) when we recited words, by establishing a novel data sampler in the LSTM model training. The dataset in the experiment is UCF101 (Soomro et al., 2012), which is a human action recognition video dataset. The name of each folder indicates the annotation of the video.

### 3.1 EBBINGHAUS FORGETTING CURVE

Ebbinghaus forgetting curve (Ebbinghaus, 1913) describes the memory effect of human brain over time, which was proposed by Hermann Ebbinghaus in 1885. This theory reveals the human memory law. It is also the law of human learning. That is, the loss of human memory when learning new knowledge is drop fast first and slow later. Ebbinghaus also pointed out that, timely review and repeated input are the key point to prevent forgetting, consolidate knowledge, and learn better. Figure 1 illustrates Ebbinghaus forgetting curve, and timely review can reduce the knowledge forgetting, which makes the learning better. Based on Ebbinghaus forgetting curve on the human brain, we simulated Ebbinghaus' method on machine learning. We believe that the experimental results in Section 4 could prove that there is a certain correlation between human learning and machine learning, since the machine learning method with timely review and spaced repeat has a faster learning effect, compared with the machine learning without the human-like method.

Ebbinghaus also found that, making good use of the correlations between knowledge is another key point for enhancing learning. We definite these correlations are temporal information in Appendix A. Thereby enhancing the use of temporal information is the key to video detection, natural language processing (NLP), etc. We believe that, the partly repeated input of the stepped sampler enhances the correlation and the temporal information.

### 3.2 LSTM

The LSTM architecture we used in this paper is to start with a CNN backbone. The CNN backbone has four convolutional layers. The dimension of convolution kernels of each convolutional layer is 32, 64, 128, 256. The size of each convolution kernel is $5 \times 5, 3 \times 3, 3 \times 3, 3 \times 3$, the stride of the convolution kernel is 2, and the padding is 0. Each convolutional layer is followed by a batch normalization (BN) (Ioffe & Szegedy, 2015) layer and a ReLU layer. The last part of the CNN model is 3 fully connected (FC) layers, which use dropout function. The dimensions of the 3 FC layers are 1024, 768 and 512 respectively.

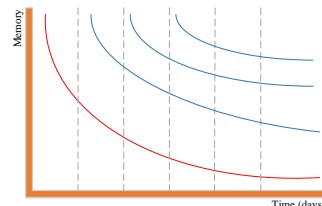

Figure 1: The illustration of Ebbinghaus forgetting curve. The red curve is without timely review (repeated input), the blue curve is with 1 day, 2 days and 3 days spaced repetition. It can be seen that, with the spaced repetition, the learning effect is better.

The LSTM used in the paper is the model existed in PyTorch. The input dimension of LSTM is 512, the hidden layer dimension is 512, and the number of hidden layers is 3. The next is two fully connected (FC) layers followed by dropout function. The dimension of the FC layers is 256. The dropout rate of the CNN backbone and LSTM are both 0.3.

### 3.3 THE STEPPED SAMPLER

Our experiment is compared with the common sampler. Common sampler in PyTorch (Paszke et al., 2019) include random sampler, weighted random sampler, batch sampler, etc. Batch sampling is nearly the most commonly used. The previous research is to add memory units to deep learning networks, such as RNN, LSTM, etc. Analogous to human learning, an important point is repetition. And the sampler should be an appropriate way to simulate the "repetition", since the data in each batch can be designed to be input repeatedly. We suppose that, "repetition" is important, not only for human beings, but also for computers. To make computers better use the "repetition", analogizing the way how we recite words, we propose a "stepped" repetition input method, which is the stepped sampler.

The structure of the proposed stepped sampler is illustrated in Figure 2. It is established on the batch sampler. The stepped sampler divides a batch into some sub-batches. Like human memory, this sampler adopts the principle of adjacent repetition (Crowder, 1968), namely, the back of the previous sub-batch is the same with the front of the next sub-batch.

The structure of the stepped sampler shows that, the input data of different batches is partly dupli-cated. The repeated input seems to increase the redundancy, but the experimental results show that, with our experimental environment, this method can accelerate the convergence of LSTM model. There is a stride between the previous sub-batch and the next sub-batch. The stride size n can be set manually. We believe that this part repetition enhances the correlation of the input frames, thereby enhancing the temporal information of the input frames, according to our definition of the temporal information in Appendix A. Section 4 describes the comparative experiments on the sampler with different stride size.

### 3.4 THE ALGORITHM OF THE STEPPED SAMPLER

The stepped sampler is designed on the basis of batch sampler. The algorithm is designed to imple-ment stepped sampling within each batch via the batch sampler. The workflow of the algorithm is as follows: the data first goes through the sequential sampler of PyTorch, then, they are processed to batches by the batch sampler. Finally, the data in each batch are divided into sub-batches with the same strides by the stepped sampler.

As shown in Figure 2, assuming that the iteration number of the stepped sampler in a batch is d, it can be concluded from the figure:

$$L = m + n \times d \tag{1}$$

It can be deduced that, the stepped sampler iteration number per batch, d is:

$$d = \frac{L - m}{n} \tag{2}$$

Equation 2 is used as the iteration number within a batch in Algorithm 1. $d$ is computed by the algorithm when $L$, $m$, $n$ is determined. If $d$ is not an integer, PyTorch will round down to ensure that $d$ is an integer. The number of batches is calculated by the framework, and the number of epochs is set manually. The algorithm of the proposed sampler is shown in Algorithm 1. The idea is to implement the stepped sampler in each batch after the sequential sampler and batch sampler of PyTorch. Line 12 of Algorithm 1 is that, after each previous stepped sub-batchs output, the starting coordinate is moved by n (step stride) data from the starting position of the previous sub-batch.

## 4 RESULTS

### 4.1 EXPERIMENT SETUP

The system used in the experiment was a workstation with 32 GB CPU RAM and a NVIDIA GeForce 1080ti GPU. The processor was Intel i7 8700, the operating system was Ubuntu 16.04 64 bits. PyTorch version used in the experiment was 1.0.1, Python version was 3.6, Numpy ver-sion was 1.20.4, Sklearn version was 0.20.4, and Matplotlib, Pandas, tqdm were implemented as the software environment. The reason why we chose old version PyTorch is, the performance of the old version may not be so powelful, however, the experimental effect may be better, since the contrast of the results may be larger. And since the old version may not have too many functions, it can focus on the factor of "repeat input", without the interferences from other irrelevant factors.

We searched the relevant literature, and found that there may be no LSTM literature that applies human learning methods to machine learning for now. Therefore, the experiment we designed is a comparison experiment, which is an ordinary CNN-LSTM, with or without the stepped sampler, and the other parameters are all the same. One of the advantages of this is that, it can reduce the influence of other irrelevant factors, and can specifically concentrate on the machine learning results of human learning methods. And the human learning methods is the timely review proposed by Ebbinghaus.

### 4.2 TRAINING

The detection accuracy evaluation and cross-entropy loss were used for the training of the models. The accuracy evaluation in the experiment used the accuracy score tool in the Sklearn package of Python. The cross-entropy loss used the function of PyTorch. The accuracy and loss were graphical-ly depicted in Figure 4, 5 and 6. The accuracy and loss were computed every epoch. The dataset of

UCF101 was split into training set and test set by a ratio of 3:1. After training, an overall accuracy and loss were computed by the test set, to evaluate the performance of the models. The epoch was set to 150. We used Adam as the optimization algorithm. We experimented different batch sizes and step sizes, which was changing the size of L, m and the step stride n shown in Figure 2.

Our experiment is trained from scratch. Training from scratch may decrease the test accuracy, but it can eliminate interference and focus on the stepped sampler. The learning rate was set to 0.0001. The momentum was set to 0.01. The operators of batch normalization (BN) and ReLU activation were used after each convolutional layer in the CNN backbone. The CNN backbone is not shown in Figure 2. The data transformation was applied to enhance the network. The input frames were transformed into $256 \times 342$ pixels.

## 4.3 EXPERIMENT RESULTS

Figure 3 are the visualized results.We tested the sampler of batch size 25. Figure 4 presents the experimental results. Each subfigure shows the training loss and test accuracy of the models. The difference of the models is only the sampler, for comparing the results only caused by the sampler. Figure 4 (a) is the model of traditional sampler, which is the sequential sampler and the batch sampler in PyTorch, and the other in Figure 4 are the models of the proposed stepped sampler. Figure 4 (b), (c), (d), (e), (f) are only different from the step stride for comparing. From Figure 4, we consider that step stride 2 (batch size 25, step size 20, in Figure 4 (c)) is the optimal. The training loss in Figure 4 (a) has many jitters, even when the epoch is more than 110, while the training loss in the other subfigures are much smoother, and can converge earlier than the traditional batch sampler model (Figure 4 (a)). Nonetheless, the test accuracy score of the traditional batch sampler model (Figure 4 (a)) is slightly higher. The test accuracy of Figure 4 (a) can be 0.656, while the test accuracy of the model with stride 2 stepped sampler (Figure 4 (c)) can be 0.603. The test accuracy of the models in Figure 4 is shown in Table 1.

From Figure 4, the following could be concluded: **a)** In the model training, LSTM with the stepped sampler converges faster than LSTM with the traditional sampler, and the convergence effect is better, i.e., there is no large jitter after the drop. **b)** When the batch size and the step size are fixed, the smaller the step stride was, the worse the detection effect became. Similarly, the larger the step stride was, the worse the detection effect became either. If the batch size and the step size are fixed, the detection effect seems to be a normal distribution of the step stride. **c)** However, LSTM with the traditional sampler whose batch size is 25 has a higher test accuracy on the test set. Although this value is not much higher than the optimal stepped sampler model (Figure 4 (c)).

From Table 1, it can be concluded that, for the same batch size, the test accuracy of the LSTM with stepped sampler rises faster than the traditional sampler LSTM. This can also be seen in Figure 4. Figure 5 and Figure 6 are the illustration of the traditional LSTM and the stepped sampler LSTM, when the batch sizes are all set to 20,15 respectively. The training loss of Figure 5 (c) and Figure 6 (c) converge faster than Figure 5 (a) and Figure 6 (a) , which denotes that, our method may have a broad-spectrum effect on machine learning. The test accuracy score of Figure 6 (c) is higher than Figure 6 (a), which denotes that, the stepped sampler LSTM may have higher test accuracy than the traditional sampler LSTM, when the batch size is 15, step size is 10, step stride is 5 of the stepped sampler LSTM, and the traditional sampler LSTM is with batch size 15. It can be seen that, there is a large jitter when the epoch is about 100 in Figure 6 (a). Figure 6 (b) and Figure 6 (c) have no large jitter after the epoch is about 60. The training loss of Figure 6 (c) drops faster than Figure 6 (a). The test accuracy of the three models is shown in Table 2. From Figure 6 we can see that, the training loss of the stepped sampler model still converges faster than the traditional sampler model.

In our experiments, most LSTM models with the stepped sampler have a more stable convergence of training loss, compared with the traditional LSTM models with the same batch size. Figure 4 (a) and Figure 6 (a) are the loss curves of the traditional batch sampler, it can be seen that, the loss curves have large jitters after the convergence. Other loss curves in Figure 4 and Figure 6 are stabler after the convergence. The stepped sampler LSTM may have a higher test accuracy than the traditional sampler LSTM, in the same batch size, which can reach the value of 0.639 (Table 2).

Our test uses the shuffle operation. ShuffleNet (Zhang et al., 2018) proves that the shuffle operation can improve the image detection mAP. We analyse that, the reason should be that the shuffle operation can reduce the correlation. According to our definition in Appendix A, the correlation

is the temporal information. Therefore, we consider that the shuffle operation can reduce the temporal information. If the shuffle operation is not used during detection, the frames are sequential. We believe that this continuity will have a certain impact on the model with temporal information. Since the test data is shuffled, there should be less temporal information among the data, the test results may reflect the detection effect better. The literature (Zhou et al., 2018) proves that, shuffle operation makes little impact on UCF101, and we think there would be no disadvantages for us to use the shuffle operation when testing.

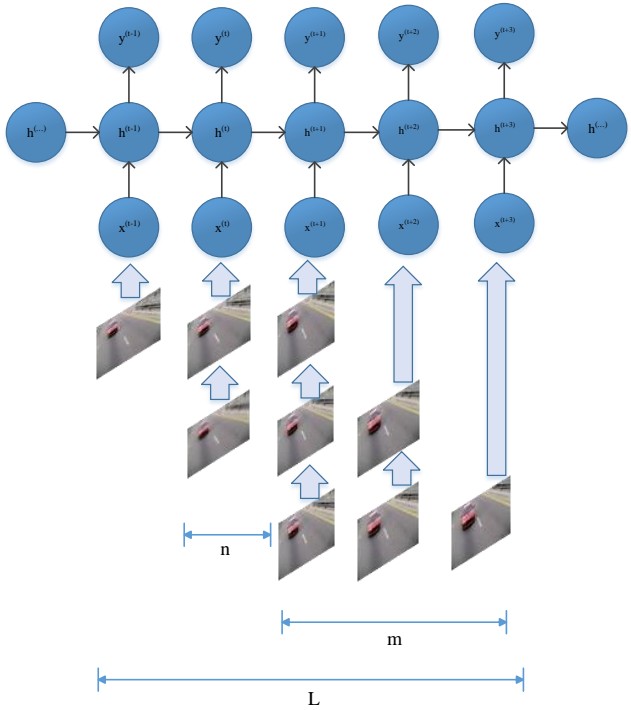

Figure 2: The illustration of the proposed stepped sampler. The upper half of the figure is the LSTM network, the lower half describes the workflow of the stepped sampler. The step size m is the size of the sub-batch, which is set to 3 in the figure, the step stride n is set to 1, the batch size L is set to 5 for the illustration. We operated the stepped sampling of the input data within each batch. The car images are from Youtube-Objects dataset (Brox & Malik, 2010).

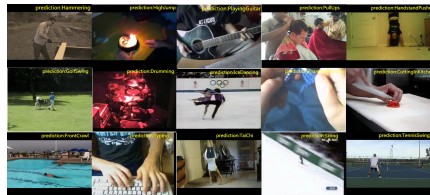

Figure 3: The video detection results of UCF101, which uses the proposed stepped sampler.

## 4.4 THE TRAINING TIME

A batch is divided into multiple sub-batches might prolong the training time. However, since the repeated data are the same, the training time of a sub-batch is much shorter than an ordinary batch. Therefore, for the total training time, the stepped sampler and the traditional sampler are almost the same. For example, when the batch sizes are all set to 15, the stepped sampler with the step stride of 5 and the traditional batch sampler both take about 60 hours for training with our experimental conditions. Moreover, we believe that the training epoch which the stepped sampler needs might be less than that of the traditional sampler.

---

**Algorithm 1:** The stepped sampler

---

**Input**: Dataset, batch size L, step size m, step stride n, and $L > m \geq n$
**Output**: Stepped sub-batch of the dataset
Initialize the dataset by Sequential sampler of PyTorch;
**for** $Batch = 1, 2, \cdots, |len(Batchsampler)|$ **do** `// use batch sampler to traverse`
`all data`
    Initialize empty set step_batch[];
    **for** $idx = 1, 2, \cdots, L$ **do**     `// traverse the elements in a batch of the`
    `batch sampler`
        output the idx-th item batch[idx] into step_batch[];
        $idx+ = 1$;
        **if** *len(step_batch[]) == m* **then** `// when the size of step_batch reaches m`
            return step_batch[];                 `// output the sub-batch`
            Reset step_batch[] to empty set;
            $idx = idx - m + n$; `// move the coordinate to the next sub-batch`
            `by stride n`
        **end**
    **end**
**end**

---

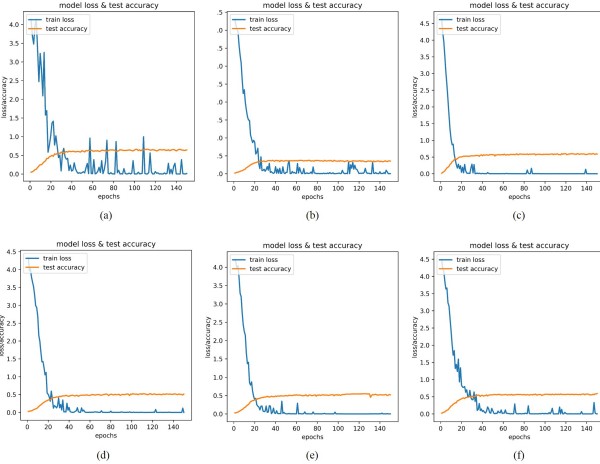

Figure 4: The training loss and test accuracy of the traditional sampler LSTM model and the stepped sampler LSTM model. The blue curves denote the training loss, and the yellow curves denote the test accuracy. The above six models are all with batch size 25. Subfigure (a) is a traditional LSTM model, which uses the batch sampler in PyTorch. Subfigure (b) is with the proposed stepped sampler, whose batch size is 25, step size is 20, step stride is 1. Subfigure (c), (d), (e), (f) are with the stepped sampler, whose batch size is 25, step size is 20, step stride is 2, 3, 4, 5, respectively. Our purpose is to study the better step stride when the batch size and step size are fixed.

Table 1: The test accuracy of the six models in Figure 4. BatchSampler denotes Figure 4(a), stride 1 stepped denotes Figure 4(b), stride 2 stepped denotes Figure 4(c), etc.

| model | epoch 10 | epoch 50 | epoch 100 | epoch 120 | epoch 150 |
|---|---|---|---|---|---|
| BatchSampler | 0.289 | 0.621 | 0.650 | 0.642 | 0.635 |
| stride 1 stepped | 0.117 | 0.369 | 0.358 | 0.360 | 0.352 |
| stride 2 stepped | 0.307 | 0.566 | 0.577 | 0.602 | 0.587 |
| stride 3 stepped | 0.133 | 0.471 | 0.506 | 0.505 | 0.514 |
| stride 4 stepped | 0.170 | 0.502 | 0.506 | 0.546 | 0.521 |
| stride 5 stepped | 0.232 | 0.553 | 0.570 | 0.568 | 0.593 |

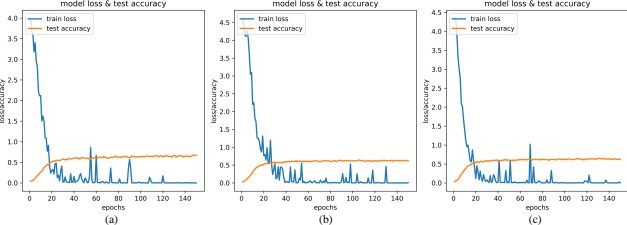

Figure 5: The training loss and the test accuracy of the traditional sampler and the stepped sampler model. The batch size of the three models are all 20. The blue curve denotes the training loss, and the yellow curve denotes the test accuracy. (a) is LSTM with the traditional batch sampler of PyTorch. (b) and (c) are LSTM with the proposed stepped sampler, using the stepped size 10, and the step stride of (b) is set to 2, and the step stride of (c) is set to 5. The training loss of (c) converges faster than (a), and the test accuracy of (c) is more stable than (a), when the test accuracy of the both seems to be equal value.

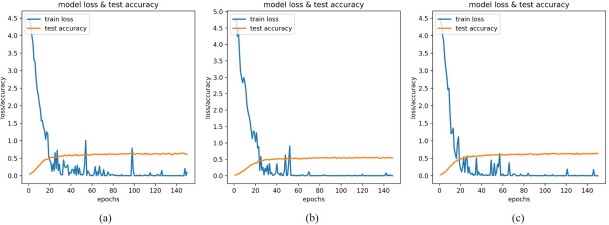

Figure 6: The training loss and the test accuracy of the traditional model and our stepped sampler model. The batch size of the three models are all 15. The blue curve denotes the training loss, and the yellow curve denotes the test accuracy. (a) is LSTM with the traditional sampler, batch sampler in PyTorch. (b) and (c) are LSTM with the proposed stepped sampler, with the same stepped size 10, when the step strides of (b) and (c) are set to 3 and 5 respectively. The training loss of (c) still converges faster than (a).

Table 2: The test accuracy of the three models in Figure 6. BatchSampler represents Figure 6(a), stride 3 stepped represents Figure 6(b), stride 5 stepped represents Figure 6(c).

| model | epoch 10 | epoch 50 | epoch 100 | epoch 120 | epoch 150 |
|---|---|---|---|---|---|
| BatchSampler | 0.303 | 0.585 | 0.607 | 0.622 | 0.616 |
| stride 3 stepped | 0.178 | 0.517 | 0.532 | 0.544 | 0.535 |
| **stride 5 stepped** | **0.282** | **0.593** | **0.631** | **0.639** | **0.637** |

# 5 DISCUSSION

The experiment is to study the detection effect of the proposed sampler, which simulates one of the human brain memory law, repeating the input, to use the temporal information (we think it is the correlation in/between frames) of videos.

As the data are partly repeated inputted, it may be equivalent to the timely knowledge review of human brain, which strengthens the memory of the LSTM network, and reduces the information forgetting. The process is very similar with human learning, which was revealed by Hermann Ebbinghaus, and it is illustrated in Subsection 3.1. LSTM can selectively memorize the temporal information, which is human-like.

From Figure 4, the repeating times of the stepped sampler is not the more the better, as shown in Figure 4 (b), the stride is 1, and the convergence speed of the model is not improved much. In addition, the repeating times of the stepped sampler is not the less the better. As shown in Figure 4 (f), the convergence speed of the model is even slower. Figure 5 (b) also seems to show this. The phenomenon is the same with human learning. Too much repetitive input and too little repetitive input would not improve the learning effect of human. The experiments seem to verify the similarity between machine learning and human learning. We assume that artificial neural networks have human-like characteristics. What is the best spaced repetition, is still need to be studied. We assume that, it should be "one solution to one issue", just like human.

Temporal information also seems to have human-like characteristics. Temporal information is the correlation of temporal sequences, analogous to human learning, it is the correlation of knowledge. In human learning, one of the important learning methods is to use the correlation of knowledge, and using the temporal information may also be one of the important learning methods of machine learning.

# 6 CONCLUSION

We refer to the human memory rules, and propose the stepped sampler, a repeating input method which uses the timely review approach. The timely review approach was proposed by Ebbinghaus, and is used to strengthen human memory and learning. In our experiments, this method has a better promotion on the detection effect of LSTM. The experimental results show that, compared with the traditional sampler, the training loss of the stepped sampler converges faster, and is more stable after the convergence, i.e., there is no large jitter after convergence. The test accuracy of the model with the stepped sampler also reaches a high point faster and is more stable either. When the batch size is 15, the test accuracy of the stepped sampler LSTM is significantly higher than that of the traditional sampler with the same batch size. We analyzed the algorithm of stepped sampler and got several equations. Ebbinghaus also pointed out that utilizing the correlations between knowledge is another key to learn better. We believe that this part repetition of the sampler enhances the correlation of the input frames, thereby enhancing the temporal information of the input frames, from our definition of the temporal information in Appendix A.

We try to use the mathematical language to describe the temporal information of video frames, which is shown in Appendix A. Since these mathematical descriptions do not involve specific artificial neural networks, the parameters of neural network are not added to the equations.

We try to use some human learning methods to study artificial neural networks. Compared with the traditional sampler, the stepped sampler LSTM has a faster learning effect, and has a higher test accuracy under certain parameters. The results show that, there may be a close relationship between biological neural network and artificial neural network, whatever in structure and even in principle. How to improve human learning is different for each individual, and the test accuracy of our experiment may illustrate this point, and we believe that is why not all the stepped sampler LSTM's test accuracy is higher than the traditional sampler. The attention mechanism (Vaswani et al., 2017) may be also inspired by human learning methods. Transfer learning (Bozinovski & Fulgosi, 1976) is using old knowledge to learn new knowledge, which may be inspired by human learning methods either. We believe that, artificial neural networks seem to have human-like characteristics, and human learning and machine learning seem to have some similarities.

ACKNOWLEDGMENTS

This work was supported in part by the National Natural Science Foundation of China under Grant 61773360.

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

# A    VIDEO TEMPORAL INFORMATION DESCRIBED IN MATHEMATICAL LANGUAGE

We consider that, video temporal information is the correlations of the objects between frames, or within a frame. We also consider that, this kind of correlation could be analogized to the correlation of human knowledge, and machine learning may be similar with human learning. From the point, we derive the equations of video temporal information from Bayes' theorem, and mutual information of information theory.

## A.1    THE TEMPORAL INFORMATION BETWEEN THE FRAMES

Bayes' theorem (Bayes & Hume, 1763), i.e. a conditional probability, can be described as:

$$P(A|B) = \frac{P(A \bigcap B)}{P(B)} \tag{3}$$

where $P(A|B)$ is the probability of event A occurring, under the condition that event B occurs. $P(A \bigcap B)$ is the probability of event A and event B occurring at the same time. $P(B)$ is the probability of event B occurring. $P(B) \neq 0$.

We consider that, Bayes' theorem reflects a temporal correlation, that is, event B occurs first, and then event A occurs. The video frame could be analogized to a kind of Venn diagram, and the objects in the frame could be analogized to the events in the Venn diagram. Thus, we apply this event temporal correlation to the temporal information between frames.

We use the area ratio of object A in the video frame R(A) to replace the occurrence probability of the event. The area ratio is defined as:

$$R(A) = \frac{area\ of\ object\ A}{area\ of\ frame} \tag{4}$$

Refer to Bayes' theorem, according to the sequential relations of the temporal information between frames, i.e., the state of the object in the next frame is derived from the state of the object in the previous frame, the temporal information between the frames of the object A is:

$$T_A(nf|pf) = \frac{R(A_{pf} \bigcap A_{nf})}{R(A_{pf})} \tag{5}$$

where "nf" represents the next frame, "pf" represents the previous frame, $T_A(nf|pf)$ denotes the temporal information between the frames of the object A, $R(A_{pf} \bigcap A_{nf})$ represents the overlapping area ratio of the previous frame and the next frame of the object A. $R(A_{pf})$ represents the area ratio of the object A in the previous frame.

Similarly, the temporal information between the frames of the object B is:

$$T_B(nf|pf) = \frac{R(B_{pf} \bigcap B_{nf})}{R(B_{pf})} \tag{6}$$

And the temporal information between the frames can be described as:

$$T_{bf} = T_A(nf|pf) + T_B(nf|pf) + \cdots = \frac{R(A_{pf} \bigcap A_{nf})}{R(A_{pf})} + \frac{R(B_{pf} \bigcap B_{nf})}{R(B_{pf})} + \cdots \tag{7}$$

where $T_{bf}$ denotes the temporal information between the frames.

## A.2    THE TEMPORAL INFORMATION WITHIN A FRAME

The temporal information within a frame reflects the correlations of objects in the same image (frame). The correlations within the same image can be represented by mutual information (Shannon, 1948) in information theory. The discrete equation of mutual information is

$$I(X;Y) = \sum_{y \in Y} \sum_{x \in X} p(x,y) log(\frac{p(x,y)}{p(x)p(y)}) \tag{8}$$

where $p(x,y)$ is the joint probability function of $X$ and $Y$, and $p(x)$ and $p(y)$ are the marginal probability functions of $X$ and $Y$.

Referring to the discrete equation of mutual information, we propose the temporal information within a frame as: when there is an overlapping area between object A and object B in the frame,

$$T(A \bigcap B) = \sum_{B \in F} \sum_{A \in F} R(A \bigcap B) log(\frac{R(A \bigcap B)}{R(A)R(B)}) \tag{9}$$

where $T(A \bigcap B)$ represents the intra-frame temporal information when object A and object B have overlapping areas, F represents all objects in the frame, $R(A \bigcap B)$ represents the overlapping area ratio of object A and object B in the frame, $R(A)$ and $R(B)$ respectively represent the area ratio of the object A and the object B in the frame.

Correlated objects in a frame generally have overlapping areas. For example, blue sky and white clouds, grass and pets, roads and cars, etc. For the correlation of objects in a frame without overlapping areas, since most of the correlated objects in a frame may have overlapping areas, the mathematical representation of the correlation of objects in a frame without overlapping areas can be weakened by a certain form.

We realize this form by taking the logarithm. Then, its mathematical expression is

$$T(A \bigcup B) = \sum_{B \in F} \sum_{A \in F} log \left( R(A \bigcup B) log(\frac{R(A \bigcup B)}{R(A)R(B)}) \right) \tag{10}$$

where $T(A \bigcup B)$ represents the temporal information in a frame with no overlapping area between object A and object B, F represents the all objects in the frame, and $R(A \bigcup B)$ represents the total area ratio of object A and object B in a frame. $R(A)$ represents the area ratio of the object A in the frame, and $R(B)$ represents the area ratio of the object B in the frame.

Since the above equation uses a logarithmic operator, when $R(A \bigcup B) log(\frac{R(A \bigcup B)}{R(A)R(B)}) < 1$, $log \left( R(A \bigcup B) log(\frac{R(A \bigcup B)}{R(A)R(B)}) \right) < 0$; when $R(A \bigcup B) log(\frac{R(A \bigcup B)}{R(A)R(B)}) > 1$, $log \left( R(A \bigcup B) log(\frac{R(A \bigcup B)}{R(A)R(B)}) \right) > 0$. This kind of positive and negative value realizes the reduction of the correlation of non-overlapping objects in a frame. Moreover, such negative values are very small. We consider that, the logarithmic operator is suitable for the correlation of non-overlapping objects in a frame.

And the temporal information within the frame can be the sum of the overlapping areas and the non-overlapping areas. It can be expressed in mathematical expression as:

$$\begin{aligned} T_{wf} &= T(A \bigcap B) + T(A \bigcup B) \\ &= \sum_{B \in F} \sum_{A \in F} R(A \bigcap B) log(\frac{R(A \bigcap B)}{R(A)R(B)}) + \sum_{B \in F} \sum_{A \in F} log \left( R(A \bigcup B) log(\frac{R(A \bigcup B)}{R(A)R(B)}) \right) \end{aligned} \tag{11}$$

where $T_{wf}$ denotes the temporal information within a frame.

## A.3  THE EQUATION OF VIDEO TEMPORAL INFORMATION

The video temporal information is the sum of the one between the frames and the one within a frame. Then, the temporal information of the frame is:

$$\begin{aligned} T &= T_{bf} + T_{wf} \\ &= \frac{R(A_{pf} \bigcap A_{nf})}{R(A_{pf})} + \frac{R(B_{pf} \bigcap B_{nf})}{R(B_{pf})} + \cdots \\ &+ \sum_{B \in F} \sum_{A \in F} R(A \bigcap B) log(\frac{R(A \bigcap B)}{R(A)R(B)}) \Big|_{A \bigcap B} \\ &+ \sum_{B \in F} \sum_{A \in F} log \left( R(A \bigcup B) log(\frac{R(A \bigcup B)}{R(A)R(B)}) \right) \Big|_{A \bigcup B} \end{aligned} \tag{12}$$

where T denotes the video temporal information. The equations we proposed try to describe the video temporal information in mathematical language. Since the analysis process is without involving the specific neural networks, the equations are only from the classic equations, without adding the parameters of optical flow and Convolutional Neural Network (CNN).

## B  THE APPLICATION OF THE EQUATIONS OF THE TEMPORAL INFORMATION IN THE EXPERIMENT

In this section, we try to apply the equation in Appendix A to analyze the experimental results in Subsection 4.3.

The test accuracy in the experiment is the detection result, and the result is of different frames sent into the model. Thus, the test accuracy can be approximately regarded as the temporal information between frames, i.e., $the\ test\ accuracy \approx T_{bf}$.

The analysis is as follows. Since the UCF101 dataset is one object in one video, Equation 7 can be transformed into $T_{bf} = T_A(nf|pf) = \frac{R(A_{pf} \bigcap A_{nf})}{R(A_{pf})}$. The test accuracy in our experiment is essentially the Intersection over Union (IoU) of the bounding boxes. Therefore,

$$The\ test\ accuracy = IoU = \frac{Area(A_{pf} \bigcap A_{nf})}{Area(A_{pf} \bigcup A_{nf})} = \frac{\frac{Area(A_{pf} \bigcap A_{nf})}{area\ of\ frame}}{\frac{Area(A_{pf} \bigcup A_{nf})}{area\ of\ frame}} = \frac{R(A_{pf} \bigcap A_{nf})}{R(A_{pf} \bigcup A_{nf})}$$
$$\approx \frac{R(A_{pf} \bigcap A_{nf})}{R(A_{pf})} = T_{bf}$$
(13)

In the above equation, since the position of the objects in the UCF101 dataset does not change much between the previous and the next frames, for the area occupied by the objects, the union set of the objects between the previous and the next frames is approximate the same as the previous frame, which can be approximately regarded as $R(A_{pf} \bigcup A_{nf}) \approx R(A_{pf})$.

## C    THE ROLE OF APPENDIX A IN THE PAPER

In Appendix A, one of our basic view points is that temporal information is a kind of correlation. In Subsection 3.1, Ebbinghaus proposed that one of the ways to improve human learning is to make good use of the correlations between knowledge. When this correlation transfers to the machine learning of video, it should be the video temporal information. Therefore, making good use of video temporal information is the key point to video detection. The sampler we proposed can enhance the temporal information.

Moreover, in Subsection 4.3, we analyzed the shuffle operation and also applied this view. Since shuffle can reduce correlations between the input data, it can also reduce the temporal information. Therefore, the interference of the temporal information to the test can also be reduced. The temporal information is undoubtedly helpful for training, since training needs the temporal information to enhance learning. However, if it is also helpful for testing, it will undoubtedly increase the test accuracy. Therefore, we added a shuffle operation to the test, for reducing the test accuracy and make the results more objective.

In Section 5, we applied this view either, and pointed out that the key of knowledge engineering lies in knowledge correlation, and for video detection and language processing, it lies in the temporal information. The above is the reason why we put forward the appendix of video temporal information.

