# OpenReview forum: "A stepped sampling method for video detection using LSTM"
_ICLR.cc/2022/Conference — ICLR 2022 Submitted_

### Official Review · Reviewer_hB9n · 2021-10-31

**Correctness:** 1
**Technical Novelty And Significance:** 1
**Empirical Novelty And Significance:** 1
**Recommendation:** 1
**Confidence:** 5

**Main Review:**

Strengths:
1, This paper gives very detailly derivation processes, equations, algorithms for the idea. The illustrated graph clearly states its idea.

Weaknesses:
1, Performances: It is useless to discuss the convergence speed without getting the optimal convergence value.  For methods comparison in RESULTS part, authors can only achieve better results when batch size equal to 15. And authors don't provide details to illustrate why choose 25 or 15 batch size. The parameters selection seems randomly. The results cannot be predicted and repeated in other situations.
2, As this paper limits its usage and performances under very narrow scope (for certain task Video detection and certain network structure LSTM) and the results are not good and seem to be unpredictable in further reproduction, there is no need to discuss other weakness. The method is useless.

**Summary Of The Paper:**

This paper mainly provides a stepped sampling method only for video detection and only adopted in LSTM structure. This idea is inspired from human's repeatedly memory working mechanism. This method can achieve fast convergence during training, smooth the training loss curves and get a good test accuracy when batch size equal to 15.

**Summary Of The Review:**

This paper proposes a very simple idea: repeatedly inputting previous frames into the same units in LSTM structure attempting to imitate humans' repeatedly memory system. But the results seem ugly, the training parameters (batch size) seem to be randomly selected. This method seems impossible to generalize to other training parameters, tasks and network structures.

---

> ### Author Response · Authors · 2021-11-21
> **Thank Reviewer's dedication for reviewing**
>
> Dear Reviewer,
>
> Thank you very much for the valuable comments. We have carefully considered your comments and addressed all of them in the paper. Our detailed responses to Reviewer’s comments can be found below.
>
> Kind regards
>
> The authors
>
> > Concern # 1: Performances: It is useless to discuss the convergence speed without getting the optimal convergence value. For methods comparison in RESULTS part, authors can only achieve better results when batch size equal to 15. And authors don't provide details to illustrate why choose 25 or 15 batch size. The parameters selection seems randomly. The results cannot be predicted and repeated in other situations.
>
>
>
> ***Author response:\*** I really apologize for our vague expression and omission very much. There are problems in our manuscript indeed. Thanks for the valuable comment. I apologize that we did not explain clearly. We repeated the experiment for many times and the results were almost similar. We also found that, for traditional sampler LSTM, the larger the batch size, the higher the test accuracy. Table 1 and Table 2 can explain this point. The reason why we chose a batch size of 15 is that, the stepped sampler's test accuracy has improved significantly (which is the highest among the specialized comparative experiment), and the second reason is that, the stepped sampler's loss curve converges faster either. The reason why we chose the batch size of 25 is that, the loss curve of the stepped sampler converges faster, and is more stable, and the second reason is that, the test accuracy of the stepped sampler is more stable either, and it is rising all the time, while the test accuracy of the traditional sampler is even fall behind.
>
> We have added the related experiments, and modified the manuscript. We thank Reviewer for pointing out the problems in our manuscript.

---

> ### Author Response · Authors · 2021-11-21
> **The authors' response (con'd) and thanks for reviewing**
>
> > Concern # 2: As this paper limits its usage and performances under very narrow scope (for certain task Video detection and certain network structure LSTM) and the results are not good and seem to be unpredictable in further reproduction, there is no need to discuss other weakness. The method is useless.
>
>
>
> ***Author response:\*** I really apologize for our vague expression very much, our manuscript has problems indeed. We have tried our best to revise the manuscript. I apologize that our manuscript has imperfections indeed. We start from Ebbinghaus forgetting curve and apply the timely review method proposed by Ebbinghaus to machine learning. Our idea is exploratory. The experimental results are in our expectation, and we believe that our study should be valuable. We have noticed that the proposal of the attention mechanism is also inspired by human learning. We want to explain an idea that human learning methods could be applied to machine learning and may have better results. We wish to be able to give our efforts some opportunities. We also look forward to seeing more human learning methods applied to machine learning. We have carefully revised all the manuscript. We thank Reviewer’s dedication for reviewing.
>
> We thank for Reviewer’s dedication, and wish that our efforts could be admitted.

---

### Official Review · Reviewer_BB9f · 2021-11-01

**Correctness:** 2
**Technical Novelty And Significance:** 2
**Empirical Novelty And Significance:** 3
**Recommendation:** 5
**Confidence:** 5

**Main Review:**

### Strengths
1. The proposed methodology is technically sound and clearly presented.
2. The experiments show promising results, lending good empirical credibility to the authors' claims.

### Weaknesses
1. My main concern is that while the authors argue that their proposed sampling technique is more "human-like" or closer to "how humans learn from data", I did not find any references in the paper backing up this claim. Are there any studies in psychology, neuroscience, or any other relevant field that grounds the key intuition for stepped sampling, which is repeating the parts of the same input in different batches? Without such grounding, the claim can come across as being made post-hoc, following the experimental success of stepped sampling.

2. Since $d$ in Eq. 11 is called the iteration number, I presumed it is a (non-negative) integer. How, then, is $d$ determined for $L=25$, $m=20$, and strides $n=2, 3, 4$ in the results shown in Figure 3?

3. The authors make claims of stepped sampling leading to stabler convergence compared to traditional sampling techniques in a few places in the paper (e.g., last paragraph in Section 4.3). Is there a way to quantify this claim, e.g., by plotting the rate of change of the loss function across the training epochs?

### Minor comments
1. Is there any specific reason the authors worked with PyTorch version 1.0.1 (Section 4.1), which came out in early 2019, when much newer versions such as 1.8.0 have been available since early 2021?

2. For the sake of completeness, please specify that $L>m \geq n$ in Algorithm 1.

**Summary Of The Paper:**

The authors present a stepped sampling to improve the learning capabilities of neural network models such as LSTM. Specifically, the stepped sampling procedure repeats the same input data in multiple batches, in other words, the batches "overlap" with one another in terms of the contained input data. This follows from the authors' argument that repeatedly providing the same data leads to faster and stable convergence in training, as well as higher accuracy in testing. The authors experimentally show these benefits of their stepped sampling procedure over traditional sampling techniques (e.g., random sampling) in the context of action detection from videos using LSTMs.

**Summary Of The Review:**

Overall, the authors propose a technically sound sampling technique and experimentally show its benefits over traditional sampling techniques, such as faster and stable convergence and higher test accuracy, in the context of action detection from videos using LSTMs. However, the authors also attribute the superiority of their technique to the claim that it is closer to how humans learn from data, but this is not grounded with any relevant prior studies or observations. Without such grounding, the claim comes off as post-hoc and the technique comes off as a successful engineering effort rather than a novel research contribution, thus making me lean towards rejection.

---

> ### Author Response · Authors · 2021-11-21
> **Thank Reviewer's dedication for reviewing**
>
> Dear Reviewer,
>
>
>
> Thank you very much for the valuable comments. We have carefully considered your comments and addressed all of them in the paper. Our detailed responses to Reviewer’s comments can be found below.
>
>
>
> Kind regards
>
> The authors
>
>
>
> > Concern # 1: My main concern is that while the authors argue that their proposed sampling technique is more "human-like" or closer to "how humans learn from data", I did not find any references in the paper backing up this claim. Are there any studies in psychology, neuroscience, or any other relevant field that grounds the key intuition for stepped sampling, which is repeating the parts of the same input in different batches? Without such grounding, the claim can come across as being made post-hoc, following the experimental success of stepped sampling.
>
> ***Author response:\*** I really apologize for our negligence, we did not explain clearly indeed. Thanks for the valuable comment. The principle of this "repeated input" sampling method is Ebbinghaus forgetting curve, which was proposed by Hermann Ebbinghaus in 1885. One of the conclusion of Ebbinghaus' theory is that, reviewing in time and repetition when learning can make learning better. We noticed that, this could be a pointcut whether there is a correlation between human learning and machine learning. Based on Ebbinghaus forgetting curve on the human brain, we simulated Ebbinghaus’ method on machine learning. We believe that the experimental results could prove that there is a certain correlation between human learning and machine learning. We have added Subsection 3.2 to the manuscript to illustrate the principle of Ebbinghaus forgetting curve we use, as follows:
>
>
>
>
>
> *3.2 Ebbinghaus Forgetting Curve*
>
> *Ebbinghaus forgetting curve (Ebbinghaus, 1913) describes the memory effect of human brain over time, which was proposed by Hermann Ebbinghaus in 1885. This theory reveals the human memory law. It is also the law of human learning. That is, the loss of human memory when learning new things is drop fast first and slow later. Ebbinghaus also pointed out that, timely review and repeated input are the key point to prevent forgetting, consolidate knowledge, and learn better. Figure 1 illustrates the forgetting curve, and timely review can reduce the knowledge forgetting. Based on Ebbinghaus forgetting curve on the human brain, we simulated Ebbinghaus’ method on machine learning. We believe that the experimental results in Section 4 could prove that there is a certain correlation between human learning and machine learning, since the machine learning method with timely review and spaced repeat has a better learning effect, compared with the machine learning without the human-like method.*

---

> > ### Comment · Reviewer_BB9f · 2021-11-27
> > **Following up on authors' responses**
> >
> > I thank the authors for carefully responding to all my questions. However, my main concern is not fully addressed for the following reasons:
> >
> > 1. While the study on human learning may be well-validated, the effects of following the same principles on machine learning still appear marginal for the single scenario the authors dealt with (video action recognition). At the very least, there needs to be convincing proof, whether mathematical or experimental, that these principles work across different network architectures and different problem scenarios.
> >
> > 2. If using a newer version of PyTorch makes the effect of stepped sampling even less prominent, does that not imply stepped sampling is doing even less of the heavy lifting?
> >
> > For these reasons, I unfortunately cannot recommend the paper for acceptance even with the current revisions.

---

> > > ### Author Response · Authors · 2021-11-28
> > > **Thank Reviewer's dedication for reviewing**
> > >
> > > Dear Reviewer,
> > >
> > > Thank you very much for the valuable comments.  Our detailed responses to Reviewer’s comments can be found below.
> > >
> > > Kind regards
> > >
> > > The authors
> > >
> > >
> > >
> > > > Concern # 1: While the study on human learning may be well-validated, the effects of following the same principles on machine learning still appear marginal for the single scenario the authors dealt with (video action recognition). At the very least, there needs to be convincing proof, whether mathematical or experimental, that these principles work across different network architectures and different problem scenarios.
> > >
> > > ***Author response:\*** We note that our conclusion is something over extended. This is why we use words "may" or "might" in the manuscript. This is one of the problems in our manuscript indeed. We'll revise our manuscript for concluding cautiously. For what Reviewer say, that is a good idea that verifying in different network structures and different application scenarios, which is a great inspiration for us. We'll try the idea in future research.
> > >
> > >
> > >
> > > > Concern # 2:  If using a newer version of PyTorch makes the effect of stepped sampling even less prominent, does that not imply stepped sampling is doing even less of the heavy lifting?
> > >
> > > ***Author response:\***  The PyTorch version is also one of the problems in our manuscript. We didn't verify the latest version of PyTorch, but just explained with our coding experience. I apologize for this. Our experiment started early, and writing codes took a long time. In addition, different versions of PyTorch library may be incompatible. Therefore, our code version is old. We'll try to use the latest version in future research.
> > >
> > > As for the role of stepped sampling, we speculate that the experimental effect may be different in different codes.
> > >
> > >
> > >
> > > We thank Reviewer's help and suggestions. We'll revise the manuscript cautiously, and actively carry out various experiments.

---

> ### Author Response · Authors · 2021-11-21
> **The authors' response (con'd) and thanks for reviewing**
>
> > Concern # 2: Since d in Eq. 11 is called the iteration number, I presumed it is a (non-negative) integer. How, then, is d determined for L=25, m=20, and strides n=2,3,4 in the results shown in Figure 3?
>
> ***Author response:\*** I really apologize for our vague expression. Thanks for the valuable comment. As Reviewer points out, d is calculated from L, m, and n. When d is not an integer, PyTorch will perform a rounding operation to ensure that d is an integer. We have already explained in Subsection 3.6, as follows:
>
>
>
> *d is computed by the algorithm when L, m, n is determined. If d is not an integer, PyTorch will round down to ensure that d is an integer.*

---

> ### Author Response · Authors · 2021-11-21
> **The authors' response (con'd) and thanks for reviewing**
>
> > Concern # 3: The authors make claims of stepped sampling leading to stabler convergence compared to traditional sampling techniques in a few places in the paper (e.g., last paragraph in Section 4.3). Is there a way to quantify this claim, e.g., by plotting the rate of change of the loss function across the training epochs?
>
> ***Author response:\*** I really apologize for our negligence. Thanks for the valuable comment. Figure 4 (a) and Figure 5 (a) are the traditional sampler LSTM, and the loss curves have large jitters after convergence. The other loss curves are the stepped sampler LSTM, and the loss curves are more stable after convergence. We have already explained in the penultimate paragraph of Subsection 4.3, as follows:
>
>
>
> *Figure 4 (a) and Figure 5 (a) are the loss curves of the traditional batch sampler, it can be seen that, the loss curves have large jitters after the convergence. Other loss curves in Figure 4 and Figure 5 are stabler after the convergence.*

---

> ### Author Response · Authors · 2021-11-21
> **The authors' response (con'd) and thanks for reviewing**
>
> > Concern # 4: Is there any specific reason the authors worked with PyTorch version 1.0.1 (Section 4.1), which came out in early 2019, when much newer versions such as 1.8.0 have been available since early 2021?
>
> ***Author response:\*** I really apologize for our negligence that we hadn’t explained. Thanks for the valuable comment. The purpose of our experiment is to verify the effect of "repeated input" on machine learning. We believe that the same version of PyTorch in the comparative experiment is sufficient for equal comparison. Moreover, because the performance of the old version of PyTorch may not be as good as the latest version, the impact on the experimental results can be seen more obvious. And because the old version has few functions, it can reduce the interference of irrelevant factors in the experiment, and can focus on the experimental results of our idea. We have already explained in subsection 4.1, as follows:
>
>
>
> *The reason why we chose old version PyTorch is, the performance of the old version may not be so powelful, however, the experimental effect may be better, since the contrast of the results may be larger. And since the old version may not have too many functions, it can focus on the factor of “repeat input”, without the interferences from other irrelevant factors.*

---

> ### Author Response · Authors · 2021-11-21
> **The authors' response (con'd) and thanks for reviewing**
>
> > Concern # 5: For the sake of completeness, please specify that L>m≥n in Algorithm 1.
>
> ***Author response:\*** I really apologize for our negligence. Thanks for the valuable comment. We have added $L>m≥n$ in Algorithm 1.
>
> We thank for Reviewer’s dedication, and wish that our efforts could be admitted.

---

### Official Review · Reviewer_JLJ8 · 2021-11-02

**Correctness:** 1
**Technical Novelty And Significance:** 1
**Empirical Novelty And Significance:** 1
**Recommendation:** 1
**Confidence:** 5

**Main Review:**

This paper presents many substantial flaws, which I detail below.

# Anachronistic temporal information representation

The authors start providing a 3 page long formulation of “video temporal information”. The authors base their formulation on the correlation of objects between frames. While this might make sense for some actions, such correlation is solely modelled as the ratio between an object bounding box and the video frame size. The authors then go on with many equations (10 of them) based on the Bayes theorem and Mutual information theory to “model video temporal information”, again based only on bounding boxes ratios.

This approach is completely oblivious of what has been done in the past 20 years to model actions. No optical flow, no visual features, no CNNs are mentioned at all. In fact, the authors do not include a single paper related to action recognition in their related work. Moreover, it is not clear how this temporal information formulation is even used, if it is used at all in this work. Indeed, after presenting their formulation, the authors continue saying they use a simple CNN with an LSTM to recognise actions. My guess is that the object-box-based formulation is not used, and that CNN features are instead employed, which leaves me to wonder why such formulation is presented in the first place.

To summarise this point, Section 3.2 (Video Temporal Information) is entirely disconnected to the rest of the paper. The temporal information formulation is overly simplistic and completely unaware of any previous work in the field. It is not clear whether this formulation is even employed.

# Insufficient contribution

The core idea of this work is to repeat frames in a batch when training an LSTM to recognise actions. This very simple idea does not constitute a sufficient contribution, because it essentially corresponds to augmenting the training data with more frames, which naturally speeds convergence. In short, no novel contribution is presented in any way since the method is just a sampler that repeats frame in a trivial way.

# Flawed and incomplete evaluation

Quoting from page 6:

> Our test uses the shuffle operation, to make the test results more objective. Since the test data is shuffled, there should be less temporal between the data, the test results may depend on the sampler and model only, which could reflect the detection effect well.

Firstly, it is unclear how shuffling would make test results more “subjective”. Secondly, it is arguably incorrect to test a model shuffling the data as this introduces noise in the evaluation. Finally and most importantly, it is not clear why the method would perform better or worse by shuffling data. This confusion stems from the fact that the authors do not specify whether they shuffle the batch (in which case shuffling makes no sense during inference) or whether they shuffle frames within a sequence (which would be a major flaw since the videos would be altered).

Besides, the method is evaluated only on a single dataset without comparison to previous work. Results show that the proposed method under-performs in many cases as well.

**Summary Of The Paper:**

This paper proposes a sampler for LSTM video models. The sampler works by repeating frames in a training batch. The method is evaluated for the task of action recognition on the UCF 101 dataset.

**Summary Of The Review:**

This paper presents major problems: it is severely unaware of previous work in the field, does not present a sufficient contribution and it is not well evaluated.

---

> ### Author Response · Authors · 2021-11-21
> **Thank Reviewer's dedication for reviewing**
>
> Dear Reviewer,
>
>
>
> Thank you very much for the valuable comments. We have carefully considered your comments and addressed all of them in the paper. Our detailed responses to Reviewer’s comments can be found below.
>
>
>
> Kind regards
>
> The authors
>
>
>
>
>
> > Concern # 1: The authors start providing a 3 page long formulation of “video temporal information”. The authors base their formulation on the correlation of objects between frames. While this might make sense for some actions, such correlation is solely modelled as the ratio between an object bounding box and the video frame size. The authors then go on with many equations (10 of them) based on the Bayes theorem and Mutual information theory to “model video temporal information”, again based only on bounding boxes ratios.
> >
> > This approach is completely oblivious of what has been done in the past 20 years to model actions. No optical flow, no visual features, no CNNs are mentioned at all.
> >
> > To summarise this point, Section 3.2 (Video Temporal Information) is entirely disconnected to the rest of the paper. The temporal information formulation is overly simplistic and completely unaware of any previous work in the field. It is not clear whether this formulation is even employed.
> >
> >
>
>
>
> ***Author response:\*** Thanks very much for the helpful comment. There are many problems in our manuscript indeed. We have modified the manuscript according to the requirements of Reviewer. I apologize for our incorrect writing. These equations are to try to explore the basic theory of computer, and try to describe the temporal information in mathematical language, without involving the specific neural networks. Therefore, we derive the equations based on the classic equations, without adding optical flow and CNN. We have moved the equations to Appendix A, and added the explanation in Appendix A, the details are as follows:
>
>
>
> *The equations we proposed below try to describe the video temporal information in mathematical language. Since the analysis process is without involving the specific neural networks, the equations are only from the classic equations, without adding the parameters of optical flow and Convolutional Neural Network (CNN).*
>
>
>
> The equation is based on the bounding box ratio indeed. The calculation of the test accuracy in the experiment is also based on the bounding box ratio. Therefore, the test accuracy value in the experiment can be used to evaluate the temporal information by the equations. The test accuracy in the experiment is essentially a kind of IoU, and it can be concluded that the test accuracy is the temporal information between frames ($T_{bf}$). We have added the content of calculating temporal information by using the test accuracy in Appendix B, as follows:
>
>
>
> *In this section, we try to apply the equation in* *Appendix A to analyze the experimental results in Subsection 4.3.*
>
>
>
> *The test accuracy in the experiment is the detection result, and the result is of different frames sent into the model. Thus, the test accuracy can be approximately regarded as the temporal information between frames, i.e., *$the\ test\ accuracy \approx T_{bf}$*. Moreover, since the UCF101 dataset is one object in one video, the Equation 5 can be transformed into *$T_{bf}=T_{A}(nf|pf)=\frac{R(A_{pf}\bigcap A_{nf})}{R(A_{pf})}$*. The test accuracy in the experiment is essentially the Intersection over Union (IoU) of the bounding boxes. Therefore,*
>
> *\begin{equation}
> \begin{aligned}
> The\ test\ accuracy &=IoU=\frac{Area(A_{pf}\bigcap A_{nf})}{Area(A_{pf}\bigcup  A_{nf})}=\frac{\frac{Area(A_{pf}\bigcap A_{nf})}{area\ of\ frame}}{\frac{Area(A_{pf}\bigcup A_{nf})}{area\ of\ frame}}=\frac{R(A_{pf}\bigcap A_{nf})}{R(A_{pf}\bigcup A_{nf})} \\ &\approx \frac{R(A_{pf}\bigcap A_{nf})}{R(A_{pf})}=T_{bf}
> \end{aligned}
> \end{equation}*
>
> *In the above equation, since the position of the objects in the UCF101 dataset does not change much between the previous and the next frames, for the area occupied by the objects, the union set of the objects between the previous and the next frames is approximate the same as the previous frame, which can be approximately regarded as *$R(A_{pf}\bigcup A_{nf})\approx R(A_{pf})$*.*
>
>
>
>
>
> I apologize that the related work is one-sided. The related work in the manuscript is mainly about sampler, because our focus is the role of sampler in machine learning. We have added the literature of action recognition in Section 2, as follows:
>
>
>
> *Muhammad et al. proposed a bi-directional long short-term memory (BiLSTM) with attention mechanism and a dilated convolutional neural network (DCNN) to perform action recognition, which outperformed the state-of-the-art methods.*

---

> > ### Comment · Reviewer_JLJ8 · 2021-11-22
> > **Quality is still not sufficient**
> >
> > I thank the authors for their effort in this rebuttal. However, I'm afraid the quality of this work is still not sufficient:
> >
> > 1. Citing only one paper to review action recognition is not enough. Not providing an adequate review of the literature for the task evaluated in this work keeps the paper in the "oblivion" state
> > 2. There is still no comparison to other works. UCF-101 is possibly the most popular dataset for action recognition, thus there are plenty of baselines one can compare against.
> > 3. Naturally, temporal continuity plays a crucial role for action recognition in videos. Depending on the actions in the dataset, shuffling frames is a potentially detrimental action. When actions are simple and can be recognised through static cues, i.e. by just looking at the background or the objects (which is the case in UCF-101), then shuffling or even dropping frames has little to no impact. When more complex actions are concerned (e.g. The Something-Something dataset) then frame shuffling is harmful. This was studied in [a] which shuffled frames in Something-Something and UCF-101 and compared classification accuracy on the shuffled and normal videos. On
> > UCF-101, where actions can be learnt via static appearance or short motion patterns, performance on the shuffled videos was virtually identical to the normal videos. On Something-Something instead results were considerably worse on the shuffled videos. To summarise on this: i) shuffling on UCF-101 is expected to have little impact, but ii) it is still a rather questionable practice to shuffle frames at test time. Once again, it is clear that this work is unaware of important past work in the field of action recognition
> > 4. I still believe using bounding boxes ratios as a base to express "temporal information" is severely limited. The many equations presented in the paper (although they were moved to the appendix) appears to me as an attempt to simply fill in some maths given the venue. The study of the temporal information as expressed in this work is brittle and irrelevant to the rest of the paper and, more broadly, to the task at hand.
> >
> > [a] Temporal Relational Reasoning in Videos. ECCV 18

---

> > > ### Author Response · Authors · 2021-11-22
> > > **Thank Reviewer's dedication for reviewing**
> > >
> > > Dear Reviewer,
> > >
> > > We thank you very much for the valuable comments and for pointing out our problems in the paper. We have carefully revised the paper. Our detailed responses to Reviewer’s comments can be found below.
> > >
> > > Kind regards
> > >
> > > The authors
> > >
> > >
> > > > Concern # 1:  Citing only one paper to review action recognition is not enough. Not providing an adequate review of the literature for the task evaluated in this work keeps the paper in the "oblivion" state
> > >
> > > ***Author response:\***  I really apologize that we only added one literature due to the limitation of the number of pages. We have added the following related literature in Section 2, as follows:
> > >
> > > *Kwon et al. (Kwon et al., 2021) proposed a spatio-temporal neighbourhood learning method on action recognition, which performed the state-of-the-art.*
> > >
> > >
> > >
> > > > Concern # 2:  There is still no comparison to other works. UCF-101 is possibly the most popular dataset for action recognition, thus there are plenty of baselines one can compare against.
> > >
> > > ***Author response:\***  I really apologize for our vague expression. Yes, our research field is action recognition. However, in this paper we want to verify our idea, which is some human learning methods can be applied to machine learning. Please forgive me for this radical view point. We are studying that how to improve our experiments. Our experiment may be one-sided, and we will carry on with the research in this area in the future.
> > >
> > >
> > > > Concern # 3:  Naturally, temporal continuity plays a crucial role for action recognition in videos. Depending on the actions in the dataset, shuffling frames is a potentially detrimental action. When actions are simple and can be recognised through static cues, i.e. by just looking at the background or the objects (which is the case in UCF-101), then shuffling or even dropping frames has little to no impact. When more complex actions are concerned (e.g. The Something-Something dataset) then frame shuffling is harmful. This was studied in [a] which shuffled frames in Something-Something and UCF-101 and compared classification accuracy on the shuffled and normal videos. On UCF-101, where actions can be learnt via static appearance or short motion patterns, performance on the shuffled videos was virtually identical to the normal videos. On Something-Something instead results were considerably worse on the shuffled videos. To summarise on this: i) shuffling on UCF-101 is expected to have little impact, but ii) it is still a rather questionable practice to shuffle frames at test time. Once again, it is clear that this work is unaware of important past work in the field of action recognition
> > >
> > > ***Author response:\*** Thanks very much for pointing out our problems and I apologize for them.  As you say, the results of shuffle can be better or worse. We learned a lot from the reference you recommend, that is, shuffle can have a worse effect on complex videos. We speculate that, this is probably because the shuffle operation reduces the temporal information of the dataset, which is our view point  in this manuscript. As you have shown, shuffling on UCF-101 is expected to have little impact, so we think there is at least no disadvantage for adding shuffle to the test. We just want to make the test accuracy of the experiment as objective as possible by adding the shuffle operation. We have added the following content in section 4.3:
> > >
> > > *The literature (Zhou et al., 2018) proves that, shuffle operation makes little impact on UCF101, and we think there would be no disadvantages for us to use the shuffle operation when testing.*
> > >
> > >
> > >
> > > Since our experiment is to verify our view point, we compare the  sampler only in the experiment. We are thinking about how to design experiments that can  verify our views more. And this is our research direction in the future.

---

> > > > ### Comment · Reviewer_JLJ8 · 2021-11-25
> > > > **Thanks, but still below acceptance threshold**
> > > >
> > > > I appreciate the authors' effort in addressing my concerns. Nevertheless, I am afraid the overall quality of this work is not good enough for ICLR, thus I keep my initial rating.

---

> > > > > ### Author Response · Authors · 2021-11-26
> > > > > **Thank Reviewer's dedication for reviewing**
> > > > >
> > > > > We thank Reviewer's help, there are problems in our manuscript indeed. We'll keep improving our manuscript and further research.

---

> > > ### Author Response · Authors · 2021-11-22
> > > **The authors' response (con'd) and thanks for reviewing**
> > >
> > > > Concern # 4:  I still believe using bounding boxes ratios as a base to express "temporal information" is severely limited. The many equations presented in the paper (although they were moved to the appendix) appears to me as an attempt to simply fill in some maths given the venue. The study of the temporal information as expressed in this work is brittle and irrelevant to the rest of the paper and, more broadly, to the task at hand.
> > >
> > > ***Author response:\*** Thanks very much for the helpful comment. I apologize for our vague writing.  We want to explain as follows:
> > >
> > > In Appendix A, one of our basic view points is that temporal information is a kind of correlation. In Subsection 3.1, Ebbinghaus proposed that one of the ways to improve human learning is to make good use of the correlations between knowledge. When this correlation transfers to the machine learning of video, it should be the video temporal information. Therefore, making good use of video temporal information is the key point to video detection. The sampler we proposed can enhance the temporal information.
> > >
> > > Moreover, in Subsection 4.3, we analyzed the shuffle operation and also applied this view. Since shuffle can reduce correlations between the input data, it can also reduce the temporal information. Therefore, the interference of the temporal information to the test can also be reduced. The temporal information is undoubtedly helpful for training, since training needs the temporal information to enhance learning. However, if it is also helpful for testing, it will undoubtedly increase the test accuracy. Therefore, we added a shuffle operation to the test, for reducing the test accuracy and make the results more objective.
> > >
> > > In Section 5, we applied this view either, and pointed out that the key of knowledge engineering lies in knowledge correlation, and for video detection and language processing, it lies in the temporal information. The above is the reason why we put forward the appendix of video temporal information. We have added the above content as Appendix C, as follows:
> > >
> > > *C THE ROLE OF APPENDIX A IN THE PAPER*
> > >
> > > **In Appendix A, one of our basic view points is that temporal information is a kind of correlation.*
> > > *In Subsection 3.1, Ebbinghaus proposed that one of the ways to improve human learning is to make good use of the correlations between knowledge. When this correlation transfers to the machine learning of video, it should be the video temporal information. Therefore, making good use of video temporal information is the key point to video detection. The sampler we proposed can enhance the temporal information.*
> > >
> > > *Moreover, in Subsection 4.3, we analyzed the shuffle operation and also applied this view. Since shuffle can reduce correlations between the input data, it can also reduce the temporal information. Therefore, the interference of the temporal information to the test can also be reduced. The temporal information is undoubtedly helpful for training, since training needs the temporal information to enhance learning. However, if it is also helpful for testing, it will undoubtedly increase the test accuracy. Therefore, we added a shuffle operation to the test, for reducing the test accuracy and make the results more objective.*
> > >
> > > *In Section 5, we applied this view either, and pointed out that the key of knowledge engineering lies in knowledge correlation, and for video detection and language processing, it lies in the temporal information. The above is the reason why we put forward the appendix of video temporal information.*
> > >
> > > We thank for Reviewer’s dedication and pointing out the problems of our paper, and wish that the efforts could be acknowledged. If have any doubts, please discuss with us.

---

> ### Author Response · Authors · 2021-11-21
> **The authors' response (con'd) and thanks for reviewing**
>
> > Concern # 2: The core idea of this work is to repeat frames in a batch when training an LSTM to recognise actions. This very simple idea does not constitute a sufficient contribution, because it essentially corresponds to augmenting the training data with more frames, which naturally speeds convergence. In short, no novel contribution is presented in any way since the method is just a sampler that repeats frame in a trivial way.
>
>
>
> ***Author response:\*** I apologize for the vagueness of our manuscript. The sampler we proposed is cyclic and stepwise, in which the data is partly repeated, and the spaced interval can be set manually. We admit that this sampler is not complicated, but we believe that this part repetition enhances the correlation of the input frames, thereby enhancing the temporal information of the input frames. We have added the content in Subsection 1.1 and Subsection 3.5, as follows:
>
>
>
>
>
> *b) A novel sampler is proposed, which implements sampling in a circular and stepwise manner. Compared with the traditional sampler, the loss curve of the LSTM using this stepped sampler in training converges faster, and is more stable after the convergence, namely there is no large jitter after the convergence. Moreover, its test accuracy curve is more stable either, which has no jitter. When the batch size is 15, the test accuracy of the stepped sampler LSTM is much higher than that of the traditional sampler with the same parameters.*
>
>
>
> *The stride size n can be set manually. We believe that this part repetition enhances the correlation of the input frames, thereby enhancing the temporal information of the input frames, according to our definition of the temporal information in Appendix A.*

---

> ### Author Response · Authors · 2021-11-21
> **The authors' response (con'd) and thanks for reviewing**
>
> > Concern # 3: Firstly, it is unclear how shuffling would make test results more “subjective”. Secondly, it is arguably incorrect to test a model shuffling the data as this introduces noise in the evaluation. Finally and most importantly, it is not clear why the method would perform better or worse by shuffling data. This confusion stems from the fact that the authors do not specify whether they shuffle the batch (in which case shuffling makes no sense during inference) or whether they shuffle frames within a sequence (which would be a major flaw since the videos would be altered).
>
>
>
>
>
> ***Author response:\*** I very apologize for our negligence that we hadn’t explained the shuffle operation. ShuffleNet significantly improves the mAP value on MS COCO by using the shuffle operation. According to our analysis, it should be that the shuffle operation can reduce the correlation. According to our definition in Appendix A, this correlation is the temporal information. Therefore, we believe that the shuffle operation can reduce the temporal information when testing, and reduce the interference of the temporal information when testing.
>
> If the shuffle operation is not performed during the test, the frames are continuous. We believe that this continuity will have a certain impact on the test results of the model with temporal information characteristic. We have added the content in the last paragraph of Subsection 4.3, as follows:
>
>
>
> *Our test uses the shuffle operation. ShuffleNet (Zhang et al., 2018) proves that the shuffle operation can improve the image detection mAP. We analyse that, the reason should be that the shuffle operation can reduce the correlation. According to our definition in Appendix A, the correlation is the temporal information. Therefore, we consider that the shuffle operation can reduce the temporal information.*
>
> *If the shuffle operation is not used during detection, the frames are sequential. We believe that this continuity will have a certain impact on the model with temporal information. Since the test data is shuffled, there should be less temporal information among the data, the test results may reflect the detection effect better.*

---

> ### Author Response · Authors · 2021-11-21
> **The authors' response (con'd) and thanks for reviewing**
>
> > Concern # 4: This paper presents major problems: it is severely unaware of previous work in the field, does not present a sufficient contribution and it is not well evaluated.
>
>
>
> ***Author response:\*** There are problems in our manuscript indeed, I apologize for our insufficient contribution very much. Our view is that human learning methods can be applied to machine learning. We have searched the literatures, but there is no relevant literature for now. Therefore, we can only compare the results with ourselves, that is, using common sampler or using the human learning-like sampler, and making contrasts. One of the goodness of this is that, because there are no other experiments, our experiments are very concentrated, only focusing on the human-like "timely review" learning method. We have added relevant content to the manuscript in Subsection 4.1, as follows:
>
>
>
> *We searched the relevant literature, and found that there may be no LSTM literature that applies human learning methods to machine learning for now. Therefore, the experiment we designed is a comparison experiment, which is an ordinary CNN-LSTM, with or without the stepped sampler. One of the advantages of this is that, it can reduce the influence of other irrelevant factors, and can specifically concentrate on the machine learning results of human learning methods. The human learning methods is the “timely review” proposed by Ebbinghaus.*
>
>
>
>
>
> We thank for Reviewer’s dedication, and wish that our efforts could be admitted.

---

### Author Response · Authors · 2021-11-21
**The revised paper is submitted and the summary of the revised points**

Dear Reviewer and Editor,

We thank Reviewer for taking time to review our manuscript. We improved the manuscript with their help.

Our initial manuscript was written in hurry. We carefully revised the manuscript according to Reviewer's comments. The following are our revise points.

1) We have moved the "video temporal information" to Appendix A, and added the explanation in Appendix A;
2) We have added the content of calculating temporal information by using the test accuracy in Appendix B;
3) We have added the literature of action recognition in Section 2;
4) We have added the subsection of "our contribution";
5) We have added the explanation of "shuffle" in the last paragraph of Subsection 4.3;
6) We have added the explanation of the experimental design in Subsection 4.1, since there is few previous work in the field, and we think it's reasonable for our experiment, which is a comparative experiment with or without the stepped sampler only, and the other parameters are all the same.
7) We have added Subsection 3.2 to the manuscript to illustrate the principle of Ebbinghaus forgetting curve we use;
8) We have added the explanation of Algorithm 1 in Subsection 3.6;
9) We have added the explanation of "stable" in the penultimate paragraph of Subsection 4.3;
10) We have added the explanation of the PyTorch version we used  in subsection 4.1;
11) We have added the condition $L>m≥n$ in Algorithm 1;
12) We have added the experiments, and revised the abstract and conclusion;
13) We have added Subsection 4.4 "The Training Time".

We have already made detailed responses to each comment, please review.

---

### Decision · Program_Chairs · 2022-01-20

**Decision:**

Reject

**Comment:**

This paper proposes a stepped sampler for LSTM-based video detection. However, reviewers raised a series of issues of this paper, including the weakness in novelty, experiment evaluations, and generalizability of the method. Considering the limited contribution of this paper, and limited experiment evaluations, the AC agrees with the reviewers and recommends reject for this paper.